# Assessment of the Main Compounds of the Lipolytic System in Treadmill Running Rats: Different Response Patterns between the Right and Left Ventricle

**DOI:** 10.3390/ijms20102556

**Published:** 2019-05-24

**Authors:** Agnieszka Mikłosz, Bartłomiej Łukaszuk, Marcin Baranowski, Adrian Chabowski, Jan Górski

**Affiliations:** 1Department of Physiology, Medical University of Białystok, 15-222 Białystok, Poland; bartlomiej.lukaszuk@umb.edu.pl (B.Ł.); marcin.baranowski@umb.edu.pl (M.B.); adrian.chabowski@umb.edu.pl (A.C.); gorski@umb.edu.pl (J.G.); 2Department of Basic Sciences, Lomza State University of Applied Sciences, 18-400 Łomża, Poland

**Keywords:** lipolityc complex (ATGL, HSL, CGI-58, G0S2), lipids (FFA, DG, TG), left and right ventricles, exercise, rat

## Abstract

The aim of the present study was to investigate the time and intensity dependent effects of exercise on the heart components of the lipolytic complex. Wistar rats ran on a treadmill with the speed of 18 m/min for 30 min (M30) or 120 min (M120) or with the speed of 28 m/min for 30 min (F30). The mRNA and protein expressions of the compounds adipose triglyceride lipase (ATGL), comparative gene identification-58 (CGI-58), G0/G1 switch gene 2 (G0S2), hormone sensitive lipase (HSL) and peroxisome proliferator-activated receptor gamma coactivator 1-alpha (PGC-1α) were examined by real-time PCR and Western blot, respectively. Lipid content of free fatty acids (FFA), diacylglycerols (DG) and triacylglycerols (TG) were estimated by gas liquid chromatography. We observed virtually no changes in the left ventricle lipid contents and only minor fluctuations in its ATGL mRNA levels. This was in contrast with its right counterpart i.e., the content of TG and DG decreased in response to both increased duration and intensity of a run. This occurred in tandem with increased mRNA expression for ATGL, CGI-58 and decreased expression of G0S2. It is concluded that exercise affects behavior of the components of the lipolytic system and the lipid content in the heart ventricles. However, changes observed in the left ventricle did not mirror those in the right one.

## 1. Introduction

It is widely known that a sedentary lifestyle and the increased consumption of a high-calorie Western diet contributes to the development of serious metabolic dysfunctions (i.e., obesity, metabolic syndrome and type 2 diabetes), which have, in the present day, reached proportions of global epidemics [1,2]. Therefore, much attention has been paid to exploring the positive role of physical activity on fatty acids metabolism. During instances of increased metabolic demand, triglycerides (TG) stored in lipid droplets of adipocytes undergo lipolysis into fatty acids (FA) and glycerol that are released into the circulation. The liberated FA provide oxidative substrates for use by different tissues including the heart [3]. Nevertheless, even up to 60% of FA entering cardiomyocytes are first esterified to TG and then directed towards oxidation [4]. This indicates that TG stored in the form of intracellular lipid droplets constitute an accessible energy reservoir but also regulate the delivery of FA to cardiomyocytes. Aside from their role as metabolic substrates, FA also activate signaling molecules such as peroxisome proliferator-activated receptors (PPARs), that in turn increase the expression of proteins involved in fat metabolism [3]. Overloading with TG leads to a reduction in the contractile function of the myocardium [5]. The myocardial TG turnover is very fast; this indicates the importance of TG as FA donors in the tissue [6]. Furthermore, the incorporation of the blood borne FA into TG protects cardiomyocytes against an excessive intracellular FA accumulation and thus prevents their toxic effects [7].

Adipose triglyceride lipase (ATGL) is the major regulator of TG lipolysis. It hydrolyzes the first ester bond of a triglyceride moiety which is the rate limiting step of the process. As a result, diglycerides (DG) and fatty acid molecules (FA) are generated. The second ester bond is hydrolyzed by hormone sensitive lipase (HSL) and monoacylglycerol (MG) is formed. Finally, MG is split into glycerol and a fatty acid by monoglyceride lipase [8]. It has also been demonstrated that ATGL activity is enhanced by the interaction with its coactivator named comparative gene identification-58 (CGI-58) [9] whereas G0/G1 switch gene 2 protein (G0S2), which binds to the patatin-like domain of ATGL, decreases its activity [10]. The important role of ATGL in TGs lipolysis is evident from observations of ATGL-deficient mice [5]. For instance, ATGL null mice showed impaired lipolysis in adipocytes and had more than 20 times greater TGs content in cardiac muscle. These 18 week old Atgl(−/−) mice exhibited typical features of congestive heart failure, i.e., marked dilatation of both left and right ventricles, congestion of pulmonary vessels and edema [5]. Additionally, exercise-induced lipolysis was also shown to be blunted in these animals [5]. Conversely, overexpression of ATGL promoted lipolysis and resulted in a reduction in the myocardial TGs level [11]. In humans, mutation in the ATGL gene caused neutral lipid storage disease with myopathy (NLSDM) accompanied by different degrees of cardiomyopathy [12].

Myocardium has been recognized as aerobic tissue using glucose (30%) and fatty acids (70%) as energy sources for ATP production. Physical exercise increases cardiac workload and thus energy requirements [13]. Therefore, exercise-induced cardiac function is accompanied by a rise in FA oxidation (since they are the major energy source for the tissue). Data related to the effect of exercise intensity on the heart TG content are controversial, with some studies showing reduced [14], whereas others showing unchanged TG volume [15]. Nevertheless, evidence indicates that exercise training affects certain components of the lipolytic system in the myocardium. For instance, ATGL expression in mice hearts was elevated after prolonged endurance training [16]. However, in the rat, the same type of training did not affect the protein level of ATGL and G0S2 but did elevate the protein expression of CGI-58 [17]. Given the importance of the discussed enzymes for lipid metabolism, it is surprising that myocardial TG, DG and FFA levels remained stable [17]. An analysis of human myocardium by magnetic resonance imaging revealed an elevated level of the heart TG after endurance training [18]. It should be stressed that the above quoted data were obtained only for the left ventricle. Recent investigations showed that tachycardia produced by electrical heart pacing reduced the mRNA expression of ATGL CGI-58, G0S2 and HSL only in the left ventricle but not in the right one [19]. Furthermore, the pacing reduced the protein expression of ATGL and G0S2 in both ventricles, but HSL protein expression decreased only in the right one [19]. However, there are no data regarding the effect of a single bout of exercise on the TG lipolytic complex in either of the ventricles.

In this study, we investigated the impact of treadmill running on the gene and protein expression of ATGL, CGI-58, G0S2 and HSL, as well as on lipids content (TG, DG and FFA) and their fatty acid composition, in the heart ventricles. Since the intensity (speed of running) and duration of exercise may induce substantial changes in the rate of lipolysis, we decided to compare the effects of both the conditions on the above-mentioned metabolic variables. For this reason, a group of rats were challenged with a single bout of treadmill run. To assess the effects of intensity of aerobic exercise (high vs. moderate) the speed of the run was set at 18 m/min (i.e., approximately 65% of the maximal oxygen uptake) and 28 m/min (approximately 82% of the maximal oxygen uptake) [20]. Regarding the duration, the rats ran for 30 min or 120 min.

## 2. Results

### 2.1. Blood Plasma FFA Content

We observed that a single bout of exercise increased total blood plasma FFA content in comparison to the control (+58%, +1.04 fold, +1.75 fold; for M30, F30, M120 vs. Ctrl; respectively; *p* < 0.05, Table 1).

Interestingly, the changes between the running groups were also noticed. The rats running for 2 h had an increased total plasma FFA content (+74% for M120 vs. M30, *p* < 0.05, Table 1). Similarly, higher intensity of the run significantly elevated, although to a smaller extent, plasma FFA concentration (+30%, F30 vs. M30, *p* < 0.05, Table 1).

### 2.2. The Left Ventricle

#### 2.2.1. The Expression of ATGL, CGI-58, G0S2 and HSL at the Post-Transcript (mRNA) Level

In comparison with the control group, we found an increased expression of mRNA for ATGL in M120 group (+83%, M120 vs. Ctrl, *p* < 0.05, Figure 1A), whereas a decrease was noticed for ATGL mRNA level in F30 (−25%, F30 vs. Ctrl, *p* < 0.05, Figure 1A). Additionally, we noticed some changes between the runs themselves, namely, increasing the duration of the run increased ATGL mRNA expression by +56% (M120 vs. M30, *p* < 0.05, Figure 1A), whereas increasing the speed decreased the mRNA expression by −36% (F30 vs. M30, *p* < 0.05, Figure 1A).

A single bout of exercise exerted its influence on the left ventricle’s CGI-58 mRNA expression compared to the control group (Figure 1B). We found a decreased expression of the mRNA in M30 group (−37%, M30 vs. Ctrl, *p* < 0.05, Figure 1B) and F30 (−25%, F30 vs. Ctrl, *p* < 0.05, Figure 1B), whereas an increment was noticed for M120 (+36%, M120 vs. Ctrl, *p* < 0.05, Figure 1B). Additionally, it appears that increasing the duration of the run from 30 min to 2 h caused a rise in CGI-58 mRNA expression by +1.15 fold (M120 vs. M30, *p* < 0.05, Figure 1B).

The animals from all the running groups were characterized by an increased expression of mRNA for G0S2 as compared with the control (+76%, +71% and +22% for M30, F30 and M120 vs. Ctrl, *p* < 0.05, Figure 1C). Additionally, we observed a time dependent effect of the run, namely the rats running for 2 h had a decreased G0S2 mRNA expression as compared with the rats running for 30 min with moderate intensity (−31% for M120 vs. M30, *p* < 0.05, Figure 1C).

The analysis of the left ventricle’s HSL mRNA expression showed no statistically significant differences between the studied groups (*p* > 0.05, Figure 1D). Despite that fact, the exercise seemed to increase HSL mRNA levels in comparison to the control, with the greatest changes observed for F30 (+37%, F30 vs. Ctrl, *p* > 0.05) and M120 (+33%, M120 vs. Ctrl, *p* > 0.05, Figure 1D). Moreover, despite not reaching a statistical significance level, it appears that increasing both the duration and the speed of the run tended to increase HSL mRNA expression (+20%, and +23%, M120 and F30 vs. M30, respectively, *p* > 0.05, Figure 1D).

#### 2.2.2. The Expression of ATGL, CGI-58, G0S2 and HSL at the Protein Level

Total protein expression of ATGL, CGI-58, and G0S2 seemed to be virtually constant among all the studied groups (*p* > 0.05, Figure 1E–G).

Similarly, the protein expression of HSL did not differ between the running groups and the control one (*p* > 0.05, Figure 1H). However, increasing the duration of the run upregulated HSL protein expression in comparison to the rats running for 30 min with moderate intensity (+21%, M120 vs. M30, *p* < 0.05, Figure 1H).

#### 2.2.3. The Expression of PGC-1α (Peroxisome Proliferator-Activated Receptor Gamma Coactivator 1-Alpha) at the Post-Transcript (mRNA) Level

In comparison with the control group, we found an increased expression of mRNA for PGC-1α in M30 and M120 group (+23% and +68%, for F30 and M120 vs. Ctrl, *p* > 0.05 and *p* < 0.05, respectively, Figure 2A).

#### 2.2.4. The Expression of PGC-1α (Peroxisome Proliferator-Activated Receptor Gamma Coactivator 1-Alpha) at the Protein Level

In comparison with the control group, all the running groups had increased PGC-1α protein expression (+16%, +13%, and +34% for M30, F30 and M120 vs. Ctrl, respectively, Figure 2B), however it reached statistical significance only in the case of M120 group (*p* < 0.05, Figure 2A).

#### 2.2.5. Lipids Content

##### Free Fatty Acids (FFA)

We noticed no differences for total FFA content as regards the comparison with the control (*p* > 0.05). Still, the greatest change was observed for 2 h run (+36% for M120 vs. Ctrl, *p* > 0.05, Figure 3A, Table A1).

Unsurprisingly, we noticed no changes in total FFA level either in response to increasing the intensity (F30 vs. M30, *p* > 0.05) or the duration (M120 vs. M30, *p* > 0.05) of the run (Figure 3A, Table A1).

##### Diacylglycerols (DG)

Only the 2 h run caused a significant drop in the total DG content (−28%, M120 vs. Ctrl, *p* < 0.05, Figure 3B, Table A2).

Interestingly, only increased duration of the run had any influence on total DG concentration in comparison with moderate intensity 30 min run. The rats from M120 group showed a decreased total DG content (−29%, M120 vs. M30, *p* < 0.05, Figure 3B).

##### Triacylglycerols (TG)

We noticed no differences in the left ventricle’s total TG content between any of the studied groups (Figure 3C, Table A3).

#### 2.2.6. Total Lipolytic Activity

We found no statistically significant changes with regard to total lipolytic activity between any of the studied groups (*p* > 0.05, Figure 4A).

### 2.3. The Right Ventricle

#### 2.3.1. The Expression of ATGL, CGI-58, G0S2 and HSL at the Post-Transcript (mRNA) Level

All the running groups had an increased expression of mRNA for ATGL as compared with the control (+69%, +1.3 fold, +74% for M30, F30 and M120 vs. Ctrl, *p* < 0.05, Figure 5A). However, we did not notice any statistically significant changes between the runs themselves.

Only the F30 group had an increased expression of CGI-58 mRNA as compared to the control (+36%, F30 vs. Ctrl, *p* < 0.05, Figure 5B). Unsurprisingly, only F30 had an increased CGI-58 mRNA level in comparison with M30 (+36%, F30 vs. M30, *p* < 0.05, Figure 5B).

We found no statistically significant differences between any of the running groups and the control animals for G0S2 (*p* > 0.05, Figure 5C). Additionally, we noticed that increasing the duration of the run from 30 min to 2 h caused a decrease in G0S2 mRNA expression by −35% (M120 vs. M30, *p* < 0.05, Figure 5C).

Regarding the right ventricle’s HSL mRNA expression, only the faster running group presented an increased expression in comparison to the control (+40%, F30 vs. Ctrl, *p* < 0.05, Figure 5D). Unsurprisingly, only this group differed from M30 with respect to the tissue HSL mRNA expression (+33% for F30 vs. M30, *p* < 0.05, Figure 5D).

#### 2.3.2. The Expression of ATGL, CGI-58, G0S2 and HSL at the Protein Level

The data obtained for the right ventricle showed no statistically significant differences between any of the investigated groups with regard to ATGL, G0S2 and HSL protein levels (*p* > 0.05, Figure 5E,G,H).

On the other hand, CGI-58 protein content did differ among the groups. In comparison with the control we found an (CGI-58) increased expression in M30 (+14%, M30 vs. Ctrl, *p* < 0.05, Figure 5F). Additionally, greater duration of the run (2 h) caused a significantly lower protein CGI-58 level in comparison with the rats running for 30 min (−15%, M120 vs. M30, *p* < 0.05, Figure 5F).

#### 2.3.3. The Expression of PGC-1α at the Post-Transcript (mRNA) Level

In comparison with the control group, we found an increased expression of mRNA for PGC-1α only in F30 group (+34% for F30 vs. Ctrl, *p* < 0.05, respectively, Figure 2C).

#### 2.3.4. The Expression of PGC-1α at the Protein Level

We noticed an increased PGC-1α protein expression in F30 group, which made it significantly different from both Ctrl and M30 animals (+15% and +23%, for F30 vs. Ctrl and M130, respectively, *p* < 0.05, Figure 2B).

#### 2.3.5. Lipids Content

##### Free Fatty Acids (FFA)

We noticed only one statistically significant difference in the total FFA level in comparison with the control (Figure 6A, Table A4). The animals running with the highest speed had a decreased right ventricular FFA content (−21%, F30 vs. Ctrl, *p* < 0.05, Figure 6A, Table A4).

Given the above, comparisons in the total FFA content between the runs showed a diminished lipid content only in the case of the fast run (−20%, F30 vs. M30, *p* < 0.05, Figure 6A, Table A4).

##### Diacylglycerols (DG)

Each exercise bout significantly decreased total DG content in comparison to the control (–16%, –33% and –17%; for M30, F30 and M120 vs. Ctrl; respectively; *p* < 0.05, Figure 6B, Table A5).

Moreover, we found some differences in the levels of total DG between the running groups themselves. Increasing the speed of the run significantly decreased total DG level as compared with moderate intensity run (−20%, F30 vs. M30, *p* < 0.05, Figure 6B, Table A5).

##### Triacylglycerols (TG)

Comparison with the control group showed that increasing the duration of run to 2 h decreased the levels of total TG (−27%, M120 vs. Ctrl, *p* < 0.05, Figure 6C, Table A6). Similarly, increasing the speed of run decreased the levels of total TG (−26%, F30 vs. Ctrl, *p* < 0.05, Figure 6C, Table A6). However, no differences in total TG contents were noticed for the rats running 30 min at the speed 18 m/min and the control group (M30 vs. Ctrl, *p* > 0.05, Figure 6C, Table A6).

Interestingly, increasing the duration as well as the intensity of the run significantly lowered total TG contents in comparison with the rats from M30 group (–37% and –35%, for M120 and F30 vs. M30, respectively, *p* < 0.05, Figure 6C, Table A6).

#### 2.3.6. Total Lipolytic Activity

In comparison with the control group, we found decreased total lipolytic activity in only the M30 group (−18%, M30 vs. Ctrl, *p* < 0.05, Figure 4B). Therefore, unsurprisingly, increasing the time and speed of the run caused an increase in the total lipolytic activity as compared with the rats running for 30 min at the speed of 18 m/min (+26% and +13%; M120 and F30 vs. M30; *p* < 0.05, Figure 4B).

## 3. Discussion

Because the heart is subjected to extraordinary metabolic challenges during physical exercise, adequate provisions of energy substrates are needed to meet the increased requirement of the working cells [21]. Interestingly, duration and intensity of exercise may evoke distinct adaptational changes in the cardiovascular system (and therefore in the heart itself). As an illustration, in the human heart an acute workload increases glucose uptake and its oxidation from both exogenous and endogenous stores [22]. On the other hand, if a moderate intensity exercise continues for a long time, the reliance on fatty acids for energy provisions dominates, as evidenced by lower values of the respiratory quotient (RQ = 0.7) [23]. Surprisingly, the data related to the effect of acute physical exercise on metabolism the heart lipids are very limited. Moreover, the research regarding the expression of main components of the lipolytic system (i.e., ATGL, G0S2, CGI-58 and HSL) at mRNA and protein level is even rarer. To the best of our knowledge, this is the first report focused on the effects of duration and intensity of a treadmill run on the above-mentioned variables in the heart ventricles. Unexpectedly, it was found that a single bout of exercise differently changed the expression of the lipolytic compounds and the lipid content in the left and right ventricles.

Our findings demonstrated that a single bout of exercise significantly elevated plasma FFA content in all of the running groups, with the highest (+1.75 fold) increment found in the case of prolonged moderately intense running (M120). Additionally, both the increase in working intensity to 28 m/min (F30 vs. M30) and duration of the exercise (M120 vs. M30) resulted in a 30% and 74% increase in plasma FFA level. These findings are in line with the work of other researchers [14] and confirm the role of plasma derived fatty acids as the major cardiac energy substrate during exercise. Previous study in Fisher-344 rats showed that the moderate intensity of exercise can almost double the body’s oxygen consumption as compared to resting state [24]. Moreover, increasing the speed of the run by 33% (18 vs. 27 m/min) increased oxygen consumption to a similar extent [24]. Therefore, there is no doubt that each exercise bout employed increased the heart workload, and as a result, more energy substrates were oxidized. Interestingly, although the plasma level of FFA was elevated, its intracellular content remained stable. This, along with a stable level of TG in the left ventricle, would suggest that during exercise plasma FFA entering the cells are directed mostly to the oxidative pathway.

### 3.1. The Left Ventricle

In our study, a single bout of exercise (treadmill running) did not affect the heart left ventricle’s TG level. To the best of our knowledge, only a few studies have examined the effects of exercise on myocardial TG content. In rats, myocardial TG content was decreased after one hour of running [14]. However, in contrary to treadmill exercise, swimming reduced TG content in the heart of rats [25]. It is likely that, the swimming exercise was more physically demanding, thus causing greater energy substrates consumption and contributing to the reduction in the myocardial TG level. In our research, however, total TG content seemed to be unaffected by any type of physical activity. Moreover, in agreement with the above, total TG lipolytic activity was also unchanged by the exercise. Our study seems to be the first that has determined total lipolytic activity in the myocardium during physical activity. Nevertheless, the available literature indicates that TG turnover in the heart is very fast and is probably the fastest of all studied tissues [6]. Additionally, increased supply of free fatty acids was shown to inhibit ATGL activity [26], thus consequently causing inhibition of endogenous of TGs’ lipolysis [27] and elevation in the myocardial TG content [28]. The discrepancy between those observations and our data may result from distinct experimental settings, namely the above-cited data were obtained under resting conditions, whereas ours at running conditions.

Surprisingly, despite the apparent stable TG content and unchanged total lipolytic activity, exercise strongly affected mRNA of certain components of the lipolytic complex. It was shown that exercise induced elevated plasma FA level may activate PPARs, that, in turn, changes cardiac metabolism [13]. However, the effect of single bout of acute exercise on PPARs expression in the heart’s left and right ventricle has not been previously examined. In our study, 120 min of moderate intense run significantly increased PGC-1α (i.e., coactivator of PPARs) expression at both transcript and protein level in the left ventricle. In contrast to the left ventricle, only the F30 group exhibited markedly elevated PGC-1α expression in the right one. Riehle et al. demonstrated that swim training in mice enhanced protein content of PGC-1α in the heart [29], whereas another study found no change in PPARα expression in mice after long-term spontaneous exercise [30]. For many years, HSL was thought to be a rate-limiting enzyme involved in the mobilization of FA from muscle and adipose tissue stores in both rest and exercise. Nowadays, however, it is known that ATGL is necessary for efficient lipolysis and other TG lipases cannot compensate for ATGL deficiency [31]. Increased duration of the exercise significantly elevated ATGL mRNA level as compared to the control and 30 min moderate intensity run. On the other hand, the fast running rats’ ATGL mRNA expression was significantly downregulated. None of these, however, were reflected in the protein content (Figure 1). It is likely that more time would be needed to translate the changes in the mRNA expression of the compounds on its protein level. Based upon this observation, one might speculate that the unchanged total lipolytic activity, together with stable TG content in left ventricle during exercise, led to the heart’s FA undersupply. However, in accordance with the importance of adipose ATGL for exercise-induced white adipose tissue (WAT) lipolysis, plasma FFA level significantly increased in all running rats. The above strongly suggests that the availability of FA was sufficient to cover energy requirements. In agreement with this notion, Schoiswohl et al. found that ATGL deficient mice exhibited a massive accumulation of TG in the heart after an acute exercise (14 m/min for 60 min, 0% grade) [32]. Despite this fact, the rodents were incapable of increasing their circulating FA levels during the exercise [32]. It is also conceivable that physical exercise increases the activities of other lipases (i.e., HSL) and the lack of changes in ATGL activity is compensated. However, in the running rats, HSL protein expression was relatively constant, despite a modest increase observed in prolonged duration moderate intensity run. HSL possesses about 10% of ATGL activity towards TGs and it is the only enzyme hydrolyzing diacylglycerols [8]. Consequently, we found that DG content was substantially diminished after prolonged running. However, studies in ATGL [33] and MGL null mice [34] suggest that HSL plays rather a minor compensatory role. These findings show that ATGL and MGL deficiency impairs lipolysis and causes a 90% drop in the acylhydrolase activity (as observed in wild animals’ adipose tissue). Moreover, this lack of ATGL and MGL activity is only partially compensated by HSL. The above is suggested by a study on HSL-deficient mice [35]. The rodents exhibited less severe lipolytic defects with respect of WAT’s lipolysis, which suggests that HSL plays a relatively minor part in the process [35]. ATGL activity is a highly regulated process with G0S2 serving as an important negative, whereas CGI-58 a positive regulator of ATGL-mediated lipolysis [9]. We found that increasing the duration of the exercise stimulates the expression of mRNA for both CGI-58 and G0S2, with the above resulting in higher ATGL transcript level (Figure 1). On the contrary, 30 min fast run significantly increased G0S2 mRNA, but diminished CGI-58 expression. Previously published data has shown that G0S2 inhibits (in a dose dependent manner) ATGL; this took place even in the presence of CGI-58 [36]. The precise molecular mechanism by which both of the proteins interact with ATGL is still elusive. However, it is known that they bind to the same N-terminal region of ATGL [37]. It is likely that in our study the opposite actions of the two proteins counterbalanced each other so that changes in the expression of ATGL protein did not reflect its total lipolytic activity. To our knowledge, no report on simultaneous elevation of the two counter-regulatory proteins has been described so far. However, the question of how exercise affects the expression of the genes of the lipolytic machinery remains open. Taken altogether, the unchanged TG content and unaffected total lipolytic activity indicate that the increased circulating FA levels themselves were sufficient to cover the increased demand for lipid utilization in working cardiomyocytes. On the other hand, it is also possible that the left ventricle of the heart was not subjected to extraordinary metabolic challenges that include pronounced increases in energy turnover during a single bout of physical exertion.

### 3.2. The Right Ventricle

Surprisingly, the results obtained for the heart’s right ventricle do not mirror those obtained for its left counterpart, despite the apparent fact that both ventricles work in concert. First of all, contrary to the left ventricle, the content of TG decreased in response to both increased duration and intensity of a run. This occurred in spite of rather stable total lipolytic activity. Interestingly, the above was due to a decline in the amount of saturated, as well as unsaturated, fatty acids composing TG fraction. Moreover, it suggested an increased utilization of myocardial TG pool in these conditions. On the other hand, TG level in the M30 group remained stable. Most likely, the working intensity (18 m/min) and/or the duration of the exercise (30 min) were too low to evoke the above-mentioned changes. Additionally, the expression of mRNA for ATGL in the right ventricle was markedly elevated after each exercise bout (Figure 5), however its protein content remained unaltered (Figure 5). The mRNA expression of ATGL activator (CGI-58) increased only in the fast speed run (Figure 5), whereas its protein content was elevated in M30 group (Figure 5). On the contrary, increasing the duration of the run decreased G0S2 (ATGL inhibitor) mRNA expression in right ventricle. Although the transcript level of HSL was increased only in the fast running group (Figure 5) diacylglycerol content was reduced after each exercise bout (Table A5, Figure 6). As indicated in the introduction to this section, changes in the response of particular components of the lipolytic complex to a single bout of exercise in the right ventricle did not follow those in the left ventricle. The reason behind the differences in the behavior of different components of the lipolytic complex between the two ventricles remain thoroughly obscure. In the dog, right ventricle coronary blood flow is around 55% of the left ventricular flow calculated per gram of the tissue [38]. At rest, oxygen extraction and oxygen consumption by right ventricle is lower than by the left one. Nonetheless, during graded exercise, oxygen extraction by the right ventricle increases whereas in left ventricle it does not due to its high extraction at rest [38]. However, so far it is uncertain whether those differences in blood flow and the degree of oxygen extraction between the ventricles may account for the difference in the main components of the lipolytic complex. It is well known that both ventricles were perfused with the arterial blood and therefore were exposed for the same blood-borne factors [39]. Moreover, it seems safe to assume that the autonomic nervous system innervating both ventricles provided the same impulses to each ventricle, enabling their coordinated work. Therefore, those factors should not be responsible for the differences in fat metabolism between the two ventricles observed in response to exercise. Cardiomyocytes, like some other cell types are equipped with a so called “mechanotransduction” mechanism. Elevation in the mechanical tension induces secretion of active compounds like mechano-growth factor (MGF) [40]. We previously hypothesized that the above-mentioned phenomenon might be accountable for the differences in the expression of the main compounds of the lipolytic complex as well as lipid metabolism between the heart’s ventricles during tachycardia [19]. Therefore, the reason behind the difference reported presently may be a consequence of so far unrevealed intrinsic factors. This interesting question warrants further very promising research.

In conclusion, we showed that exercise affects mRNA but not protein expression of the principle compounds of the lipolytic complex in the rat heart ventricles and that the alterations are time and intensity dependent. However, the changes observed in the left ventricle did not mirror those in its right counterpart. We can speculate that during running, increased concentration of plasma FFA was sufficient to maintain adequate provisions of energy substrates. In that case, only relatively insignificant portions of ATP were obtained from intramyocellular TG/FFA stores. Nevertheless, we also observed the existence of several differences in the mRNA expression of ATGL, its activator (CGI-58), its inhibitor (G0S2) and HSL between the ventricles. The exercise bouts did not affect the examined lipid level in the left ventricle. However, the reductions in their levels occurred in the right ventricle. The right ventricle muscle is much thinner and develops much lower systolic pressure than its left counterpart [39]. Therefore, we may assume the existence of different working conditions in each ventricle. Therefore, we may hypothesize the existence of different local factors specific to a given ventricle. Their actions might explain the differences in behavior of the examined lipolytic system compounds and fat metabolism between the ventricles. We suggest a possible role of intrinsic factors in the regulation of endogenous TG lipolysis in each ventricle. The nature of those factors remains unrecognized, so far.

## 4. Material and Methods

### 4.1. Animal Experiments

Forty male Wistar rats (age: eight weeks, body weight: 261 ± 5 g) were used in this study. The animals were housed in a temperature-controlled room (22 ± 1 °C) with a 12 h reversed light–dark cycle. Pellet chow for rodents and tap water were provided during the experimental period. All procedures were approved by the Ethical Committee for Animal Experiments at the Medical University of Bialystok, Poland (permission no.: 72M/2017, date: 31 October 2017). The rats were randomly divided into four groups (*n* = 10 per group):Control (group designation: Ctrl),Moderately intense run on an electrically driven treadmill with the speed of 18 m/min for 30 min (group designation: M30),Fast (high intensity) run on an electrically driven treadmill with the speed of 28 m/min for 30 min (group designation: F30),Moderately intense run on an electrically driven treadmill with the speed of 18 m/min for 120 min (group designation: M120).

The treadmill was set at 10° incline. All animals first performed a preliminary treadmill adjustment by running 15 min daily for five days with the same speed as in the proper experiment. Prior to sacrifice, the rats were anesthetized with pentobarbital administered by intraperitoneal injection in a dose of 80 mg/kg of body weight. Blood samples were obtained from the abdominal aorta and centrifuged for 10 min at 4000 r.p.m. The samples were taken from right and left ventricle, cleaned from blood, promptly frozen in liquid nitrogen and stored at −80 °C until final examinations.

### 4.2. Lipid Content

Tissue (TG, DG and FFA) and plasma lipids (FFA) were analyzed by gas liquid chromatography as described previously [41,42]. Briefly, the frozen muscle samples were pulverized in an aluminum mortar precooled in liquid nitrogen. The heart tissues (20 mg) as well as blood plasma (200 µL) samples were extracted with a chloroform-methanol (2:1 vol/vol) solution with an addition of antioxidant (0.01% butylated hydroxytoluene). Next, an internal standard (100 μL) containing heptadecanoic acid (C17:0 FFA), 1,2-diheptadecanoin (C17:0 DG) and triheptadecanoin (C17:0 TGs) (Sigma-Aldrich, St. Louis, MO, USA) was added. After overnight extraction, the samples were separated using TLC on silica gel plates (Silica Plate 60, 0.25 mm; Merck, Darmstadt, Germany) with a heptane: isopropyl: acetic acid (60:40:3, vol/vol/vol) resolving solution. After visualization the lipid bands containing TG, DG and FFA were scraped off and methylated. The fatty acid methyl esters (FAMEs) were extracted using pentane. Thereafter, the samples were dissolved in hexane and analyzed using a Hewlett-Packard 5890 Series II gas chromatograph with Varian CP-SIL88 capillary column (50 mm × 0.25 mm internal diameter) and a flame-ionization detector (FID) (Agilent Technologies, Santa Clara, CA, USA). The individual fatty acids were quantified according to the standards retention times. Total plasma FFA, and the muscle tissue TGs, DG and FFA concentrations were estimated as the sum of particular fatty acid species contents in the assessed fraction. The value was expressed as nanomoles per milliliter in blood plasma and as nanomoles per gram of wet tissue in the muscles.

### 4.3. Total Lipolytic Activity

The total lipolytic activity in the muscle samples was measured using Lipase Activity Assay Kit (Sigma-Aldrich), incorporating triacylglycerol as a substrate, according to the manufacturer’s instruction.

### 4.4. Quantitative Real-Time PCR Analysis

Total RNA was isolated from the rats’ ventricles using the NucleoSpin RNA Plus Kit according to the manufacturer’s protocol (Macherey Nagel GmbH & Co.KG, Duren, Germany). The RNA was subsetuently treated with RNase-free DNase I (Ambion, Thermo Fisher Scientific, Waltham, MA, USA) and quantified by spectrophotometry. The synthesis of the complementary DNA was done using the EvoScript universal cDNA master kit (Roche Molecular Systems, Boston, MA, USA). Whereas, quantitative real time polymerase chain reaction (qRT-PCR) was carried out using the LightCycler 96 System Real-Time thermal cycler with FastStart essential DNA green master (Roche Molecular Systems). Cycling conditions were: 15 s denaturation at 94 °C, 30 s annealing at 60 °C for Cyclophilin A, PGC1α, 61 °C for ATGL, HSL, G0S2 and 62 °C for CGI-58 and 30 s extension at 72 °C for 45cycles. Primer sequences used in this study:Cyclophilin A—F:5′-TGTCTCTTTTCGCCGCTTGCTG-3′; R:5′-CACCACCCTGGCACATGAATCC-3′;ATGL—F:5′-CCCTGACTCGAGTTTCGGAT-3′; R:5′-CACATAGCGCACCCCTTGAA-3′;G0S2—F:5′-TGACCTCCTTCAGCGAGTG-3′; R:5′-TCGGGACTTCTGCGTCATC-3′;CGI-58—F:5′-AACCCCAAGTGGTGAGACAG-3′; R:5′-GCGCCGAAGATGACTGAAAC-3′;HSL –F:5′-AATGACACAGTCGCTGGTGGCG-3′; R:5′-TGCCACACCCAAGAGCTGACCT-3′PGC1α—F: 5′-ACAGACACCGCACACATCGC-3′; R: 5′-GCTTCATAGCTGTCATACCTGGGC-3′;

Melting curve analysis was performed at the end of each reaction to verify PCR product specificity. Expression of each gene was quantified by measuring Ct values, normalized to Cyclophilin A and calculated according to Pfaffl method [43]. Finally, the control was set to 100 and the experimental groups were expressed relatively to the control. All the samples were assayed in duplicate.

### 4.5. Immunoblotting

Western blotting procedures were used to assess protein content as previously described [2,44]. Briefly, the samples were homogenized in an ice-cold RIPA buffer containing a cocktail of protease and phosphatase inhibitors (Roche Diagnostics GmbH, Mannheim, Germany). The protein concentration was determined using the BCA method with bovine serum albumin (BSA) as a standard. Next, homogenates (40 μg of the total protein) were reconstituted in Laemmli buffer, separated with sodium dodecyl sulfate-polyacrylamide gel electrophoresis and transferred to a polyvinylidene difluoride (PVDF) membranes. The PVDF membranes were incubated overnight with antibodies i.e., ATGL (1:1000, cat. no. ab109251, Abcam, Cambridge, UK), HSL (1:1000, cat. no. NBP1-00879, Novus Biologicals, Centennial, CO, USA), G0S2 (1:1000, cat. no. Ab183465, Abcam), GAPDH (1:1000, cat. no. ab9385, Abcam) and CGI-58 (1:500, cat. no. NB110-41576, Novus Biologicals), PGC1α (1:1000, cat. no. NBP1-04676, Novus Biologicals). Thereafter, the PVDF membranes were incubated with the appropriate secondary antibodies conjugated with horseradish peroxidase (Santa Cruz Biotechnology, Dallas, TX, USA). Protein bands were visualized using an enhanced chemiluminescence substrate (Thermo Scientific, Rockford, IL, USA) and quantified densitometrically (ChemiDoc visualization system EQ, Biorad, Warsaw, Poland). Equal protein concentrations were loaded in each lane, which was confirmed by Ponceau S staining. Protein expression (Optical Density Arbitrary Units) was normalized to GAPDH. Finally, the control was set to 100 and the experimental groups were expressed relatively to the control.

### 4.6. Statistics

The obtained results were analyzed using *R*, a statistical software/programming language (https://cran.r-project.org/, version 3.4.4). Briefly, in the first step, the assumptions of the statistical methods were checked. Shapiro-Wilk and Fligner-Killeen tests were employed to test the normality of distributions and homogeneity of variances, respectively. Next, the data that positively passed the tests were analyzed using analysis of variance (ANOVA) followed by post-hoc pairwise Student’s *t*-tests. Whenever the assumptions of the parametric methods did not hold Kruskal-Wallis rank test with the subsequent pairwise, Wilcoxon tests were applied. False discovery rate was controlled by adjusting the obtained *p*-values using Benjamini-Hochberg correction. The corrected *p*-values that fell below the cutoff level of 0.05 were considered to be statistically significant. For consistency, all the data have been presented as mean and standard error of the mean (SEM).

## Figures and Tables

**Figure 1 ijms-20-02556-f001:**
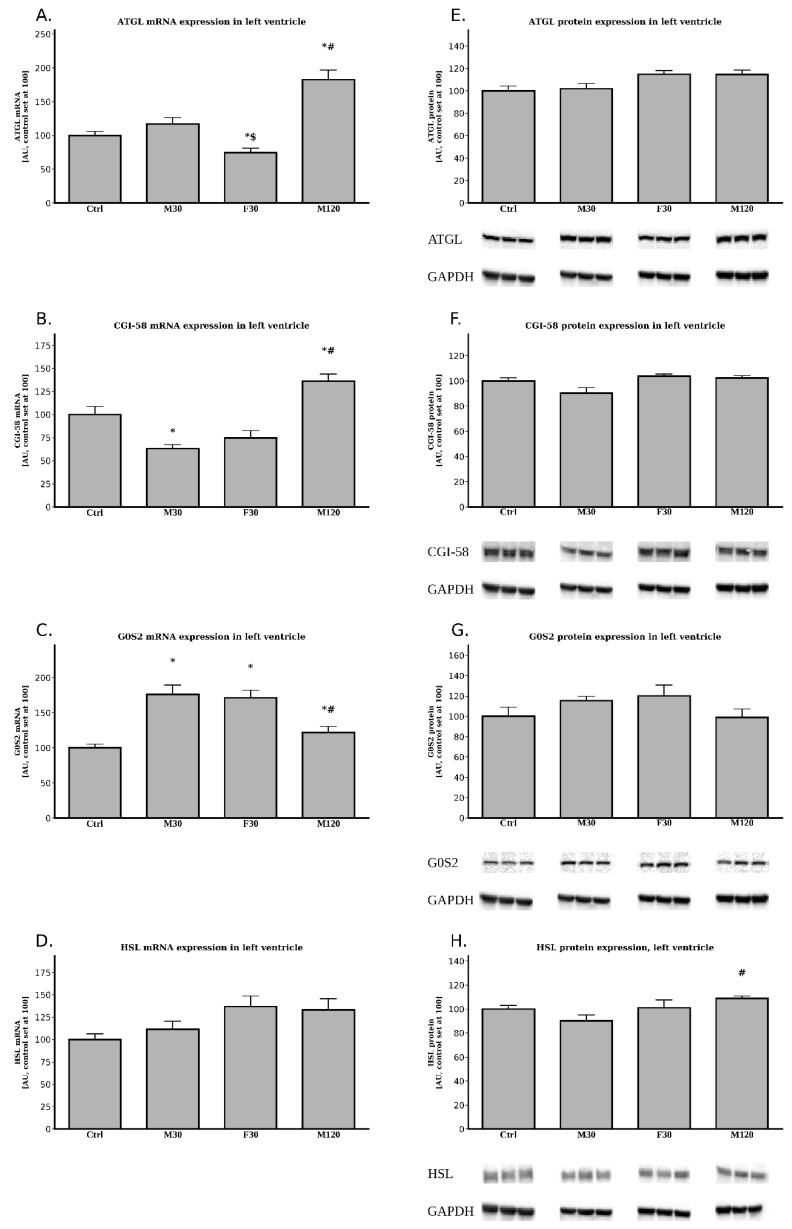
Effects of treadmill running on the mRNA and protein expressions of ATGL (**A**,**E**), CGI-58 (**B**,**F**), G0S2 (**C**,**G**), HSL (**D**,**H**) in the left ventricle. Data are expressed as mean ± SEM. For the sake of clarity, the control group was set at 100, and exercised groups were scaled with respect to Ctrl * *p*  <  0.05 difference vs. control (Ctrl); ^#^
*p*  <  0.05 difference M120 vs. M30; ^$^
*p* < 0.05 difference F30 vs. M30. Adipose triglyceride lipase (ATGL), comparative gene identification-58 (CGI-58), G0/G1 switch gene 2 (G0S2), hormone sensitive lipase (HSL).

**Figure 2 ijms-20-02556-f002:**
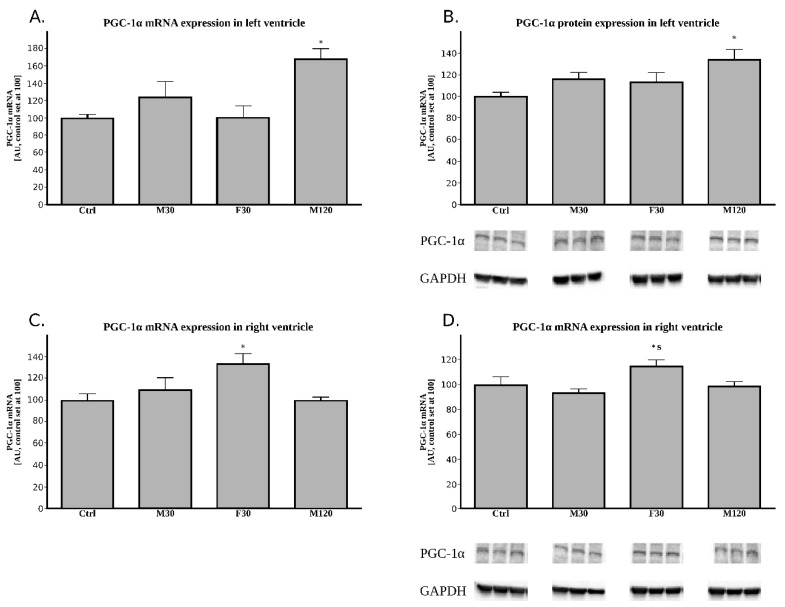
Effects of treadmill running on the peroxisome proliferator-activated receptor gamma coactivator 1-alpha (PGC-1α) tissue expression. (**A**) PGC-1α mRNA expression in left ventricle, (**B**) PGC-1α protein expression in left ventricle, (**C**) PGC-1α mRNA expression in right ventricle, (**D**) PGC-1α protein expression in right ventricle. Data are expressed as mean ± SEM. For the sake of clarity, the control group was set at 100, and exercised groups were scaled with respect to Ctrl * *p* < 0.05 difference vs. control (Ctrl); ^$^
*p* < 0.05 difference F30 vs. M30. *n* = 5 (per group).

**Figure 3 ijms-20-02556-f003:**
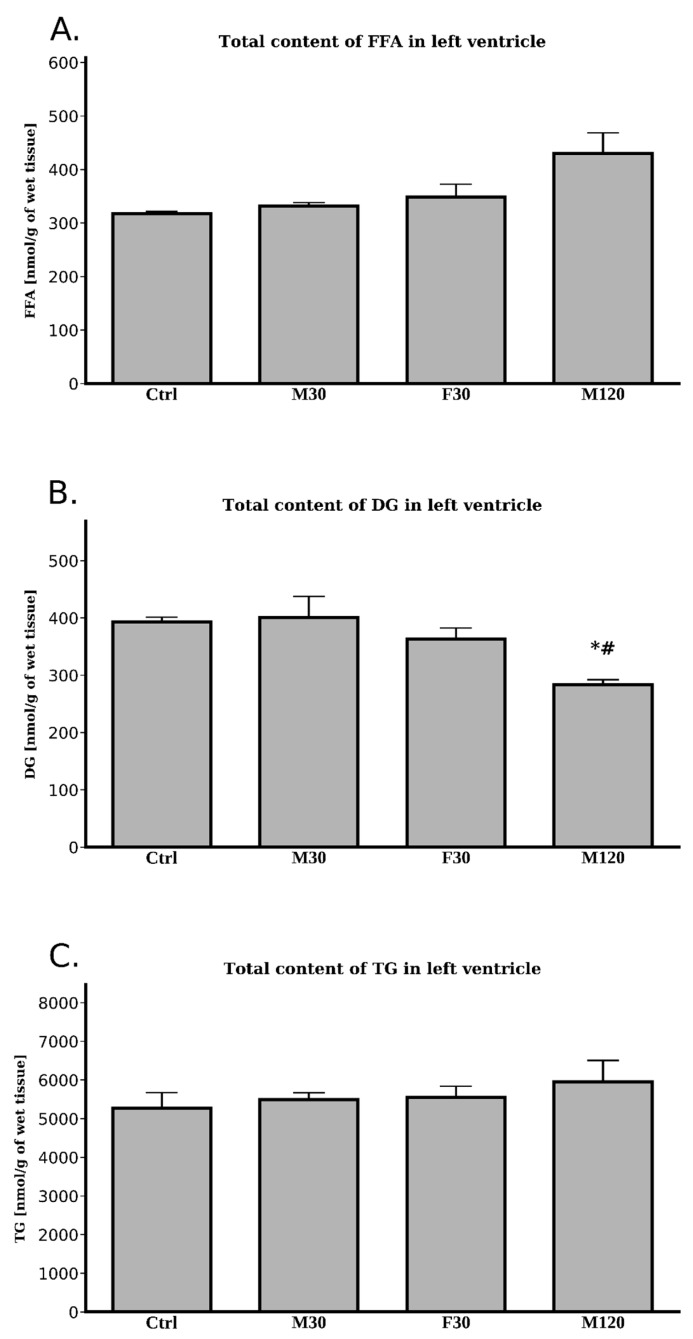
Effects of treadmill running on the total lipids content: FFA (**A**) DG (**B**) TG (**C**) in the left ventricle. Data are expressed as mean ± SEM. * *p*  <  0.05 difference vs. control (Ctrl); ^#^
*p*  <  0.05 difference M120 vs. M30. Diacylglycerol (DG), triacylglycerol (TC).

**Figure 4 ijms-20-02556-f004:**
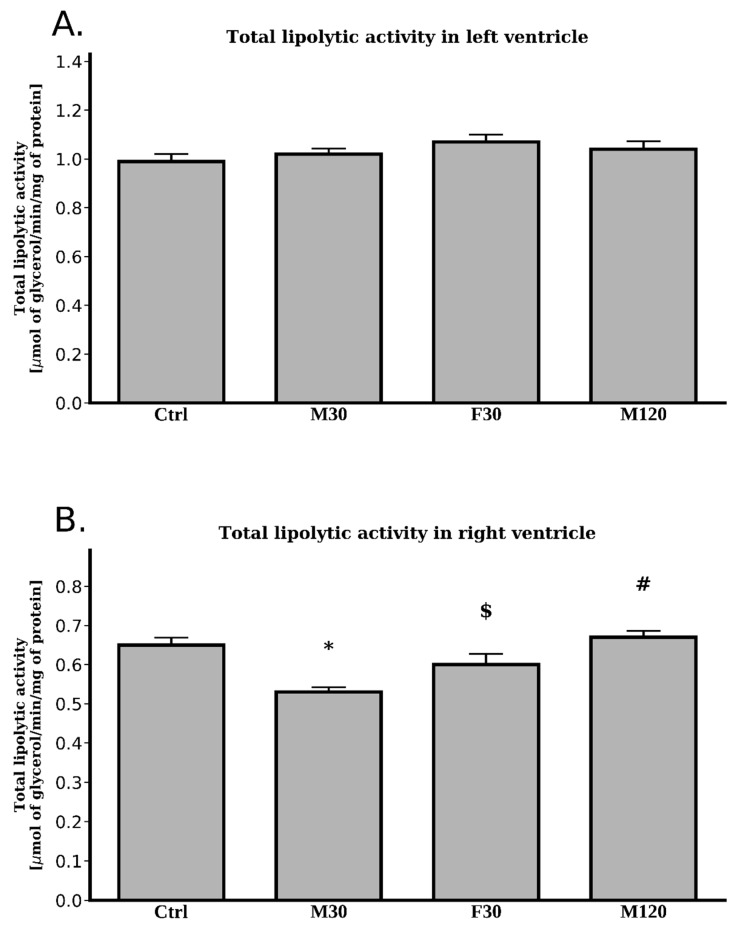
Effects of treadmill running on the total lipolytic activity in left ventricle (**A**), right ventricle (**B**). Data are expressed as mean ± SEM. * *p*  <  0.05 difference vs. control (Ctrl); ^#^
*p*  <  0.05 difference M120 vs. M30; ^$^
*p* < 0.05 difference F30 vs. M30.

**Figure 5 ijms-20-02556-f005:**
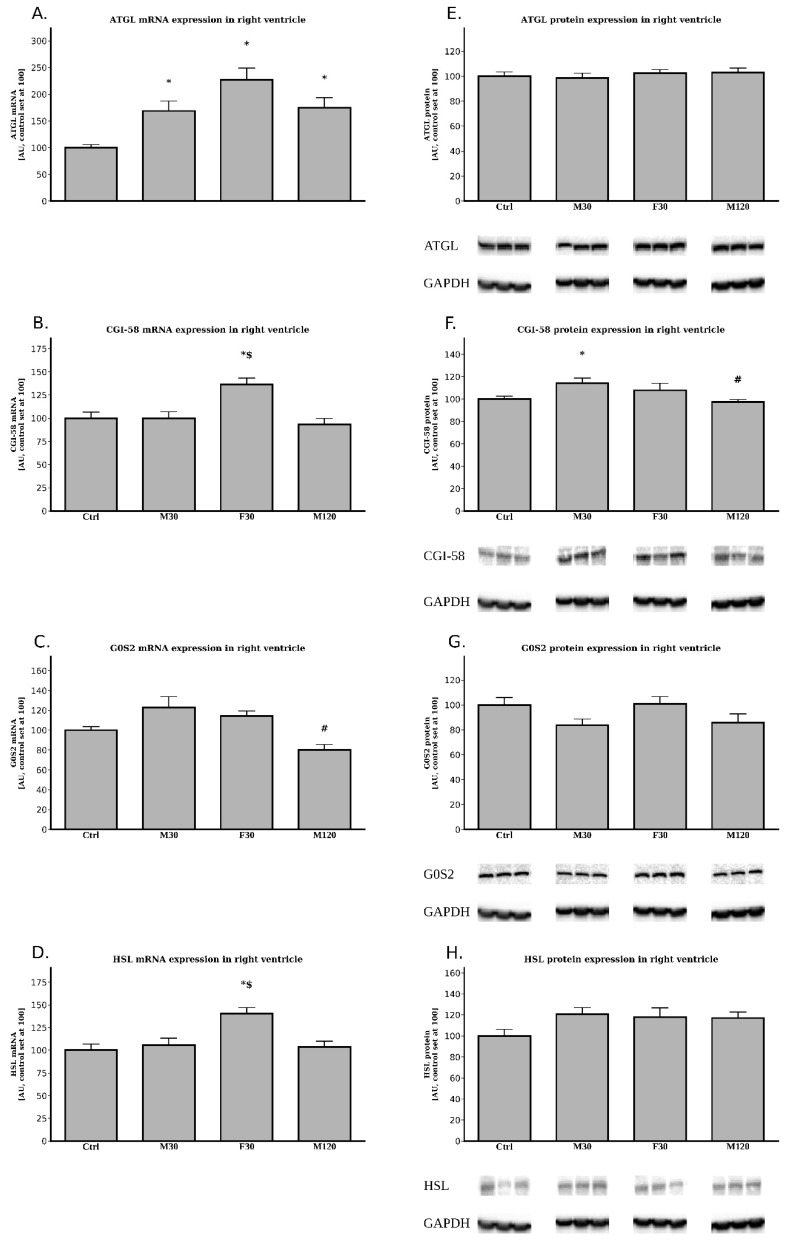
Effects of treadmill running on the mRNA and protein expressions of ATGL (**A**,**E**), CGI-58 (**B**,**F**), G0S2 (**C**,**G**), HSL (**D**,**H**) in the right ventricle. Data are expressed as mean ± SEM. For the sake of clarity, the control group was set at 100, and exercised groups were scaled with respect to Ctrl * *p*  <  0.05 difference vs. control (Ctrl); ^#^
*p*  <  0.05 difference M120 vs. M30; ^$^
*p* < 0.05 difference F30 vs. M30.

**Figure 6 ijms-20-02556-f006:**
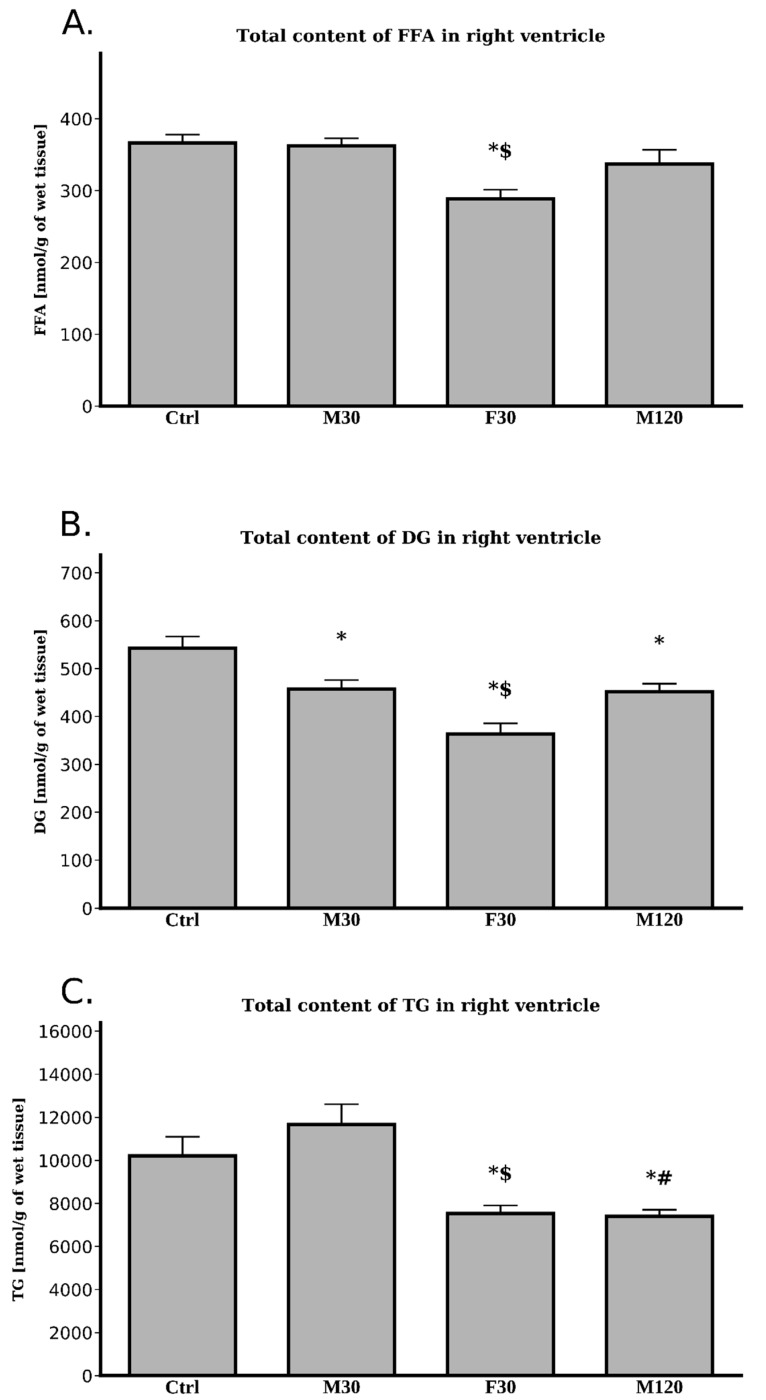
Effects of treadmill running on the total lipids content: FFA—free fatty acids (**A**) DG—diacylglycerol (**B**) TG—triacylglycerol (**C**) in the right ventricle. Data are expressed as mean ± SEM. * *p*  <  0.05 difference vs. control (Ctrl); _#_
*p*  <  0.05 difference M120 vs. M30; ^$^
*p* < 0.05 difference F30 vs. M30.

**Table 1 ijms-20-02556-t001:** Plasma fatty acid composition (FFA) (nmol/mL of plasma).

Fatty Acid	Ctrl	M30	F30	M120
Myristic acid (14:0)	4.69 ± 0.51	2.92 ± 0.3 *	7.15 ± 0.717 *^$^	10.02 ± 0.52 *^#^
Palmitic acid (16:0)	52.06 ± 4.233	81.94 ± 6.083 *	111.14 ± 3.421 *^$^	138.12 ± 5.074 *^#^
Palmitooleic acid (16:1)	4.38 ± 0.717	9.75 ± 1.163 *	16.41 ± 1.233 *^$^	19.49 ± 1.505 *^#^
Stearic acid (18:0)	13.65 ± 0.596	18.18 ± 1.063 *	19.42 ± 0.522*	24.59 ± 0.906 *^#^
Oleic acid (18:1n9c)	19.34 ± 1.98	34.08 ± 2.999 *	44.35 ± 1.308 *^$^	44.79 ± 3.132 *^#^
Linoleic acid (18:2n6c)	42.57 ± 4.44	67.27 ± 5.322 *	83.77 ± 3.798 *^$^	99.7 ± 5.476 *^#^
Arachidic acid (20:0)	0.37 ± 0.017	0.33 ± 0.015	0.32 ± 0.02	44.5 ± 9.838
Linolenic acid (18:9n3)	4.87 ± 0.617	9.14 ± 0.737 *	11.38 ± 0.578 *^$^	10.33 ± 1.167 *
Behenic acid (22:0)	0.31 ± 0.019	0.32 ± 0.019	0.33 ± 0.019	5.96 ± 1.25 *^#^
Arachidonic acid (20:4n6)	6.28 ± 0.474	10.01 ± 0.644 *	10.03 ± 0.267 *	9.07 ± 0.965 *
Lignoceric acid (24:0)	0.44 ± 0.026	0.44 ± 0.031	0.45 ± 0.051	4.72 ± 1.012
Eicosapentaenoic acid (20:5n3)	0.26 ± 0.026	0.64 ± 0.051 *	0.66 ± 0.044 *	0.72 ± 0.068 *
Nervonic acid (24:1)	0.15 ± 0.011	0.16 ± 0.009	0.14 ± 0.009	0.44 ± 0.061 *^#^
Docosahexaenoic acid (22:6n3)	1.46 ± 0.156	2.42 ± 0.2 *	2.2 ± 0.133 *	1.6 ± 0.192 ^#^
SAT	71.51 ± 5.299	104.14 ± 7.298 *	138.81 ± 4.176 *^$^	227.92 ± 17.043 *^#^
UNSAT	79.32 ± 8.008	133.47 ± 10.703 *	168.95 ± 6.138 *^$^	186.14 ± 9.822 *^#^
Total	150.83 ± 13.165	237.61 ± 17.776 *	307.77 ± 9.524 *^$^	414.06 ± 14.435 *^#^

Effects of treadmill running on plasma free fatty acids content and FA composition (mean  ±  SEM). * *p*  <  0.05 difference vs. control (Ctrl); ^#^
*p*  <  0.05 difference M120 vs. M30; ^$^
*p* < 0.05 difference F30 vs. M30. Control (Ctrl), moderately intense run (M30) (speed: 18 m/min, duration: 30 min), moderately intense run (M120) (speed: 18 m/min, duration: 120 min), fast run (F30) (speed: 28 m/min, duration: 30 min). Free fatty acid (FFA).

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
