# Peer review of "Assessment of the Main Compounds of the Lipolytic System in Treadmill Running Rats: Different Response Patterns between the Right and Left Ventricle"

_ijms, 2019, doi:10.3390/ijms20102556_

Round 1
Reviewer 1 Report
Agnieszka et al checked the effect of one time exercise on lipid content of heart components. They checked the FFA, TG,DG, signaling molecules both at mRNA and protein level. They introduced the background of study very well. They found very different effects of exercise on lipids of left and right ventricles which is novel part of this study. Following are the some question author should explain.
1. All rats were pre-trained over treadmill for 15 minutes daily for five days . I think this training itself can cause changes in lipid content and signaling in heart and plasma. How did you nullify this effect in your study?
2. ATGL level should increase after any exercise but you found 30 minutes (at speed of 28 m/min) of exercise causes decrease of ATGL in left ventricle of heart. How is it possible if FFA are increasing but ATGL expression level is decreasing?
3. CGI-58 is positive regulator of ATGL and lipolysis. You found CGI-58 mRNA is decreasing in left ventricles after 30 minute moderate and intense exercise. How is it possible?
4. Similarly you found increase in mRNA level of G0S2 which is negative regulator of lipolysis. Can you give explanation for that?
5. You found increase in FFA level in plasma but not in left ventricles of heart, does it mean heart tissues keep using glucose as a source of energy after exercise?
6. ‘Two hours run causes a significant drop in the total DG content’ (line 154). How is it possible if total FFA level is not changing?
7. Why did only F30 group show increase CGI-58 expression in right ventricle but no other exercise group?
8. Why did long exercise cause (120 minutes) decrease in CGI-58 protein level? Should it not increase?
9. 'General schema of the conducted research' (line 379) is hard to understand and it is not fully explanatory. Either remove it or make more explanatory.
10. Why treadmill was inclined at 100 ?
11. PCR condition is not described in method.
12. Catalogue number for antibodies is missing in method section.
Author Response
Dear Reviewer 1,
Thank You for Your time, effort and comments that have contributed to the improvement of our manuscript. They are all much appreciated. Below You will find our answers. For Your convenience we have placed all the changes using red font. We hope that You will find them more than satisfactory.
Best regards, Agnieszka Miklosz – corresponding author
1. All rats were pre-trained over treadmill for 15 minutes daily for five days. I think this training itself can cause changes in lipid content and signaling in heart and plasma. How did you nullify this effect in your study?
Thank You for the comment. Actually, all animals (also those form the control group) underwent a preliminary treadmill adjustment by running 15 min daily for five days with the same speed as in the proper experiment. Of course, You are right that such training itself can cause changes in lipids metabolism in heart and in plasma. However, since all rats were familiarized with the treadmill in the same manner, we believe it should ‘nullify’ this effect in our study. Moreover, we would like to stress out that a preliminary treadmill adjustment that was performed is in line with the those found in the literature.
2. ATGL level should increase after any exercise but you found 30 minutes (at speed of 28 m/min) of exercise causes decrease of ATGL in left ventricle of heart. How is it possible if FFA are increasing but ATGL expression level is decreasing?
As one may expect, there is a commonly observed significant increase in serum FFA due to systemic activation of ATGL (mainly in WAT). In line with that notion (ATGL induced WAT lipolysis in the exercised animals) also in our study the levels of circulating FFA significantly increased in all the running rats. One could also expect tissue specific activation of ATGL (the heart). This seem not to have happened in our study since we noticed unchanged lipids content in the left ventricle after acute treadmill run. We are not entirely sure of the background causes ourselves (this is an initial study on the subject). Perhaps, the intracellular FFA level isn’t a factor affecting ATGL mRNA expression in the heart or it requires a longer time frame in order to be observed. The obtained data, however, appear to be consistent and suggest another possibility. All the exercised rats exhibited a stable total lipolytic activity in the left ventricle. We did observe significantly lower mRNA expression for ATGL in the F30 group, however its protein content was even slightly elevated (+15%, F30 vs. Ctrl, p>0.05). In line with the reduced mRNA expression of ATGL in F30 group stays the fact of decreased transcript level of its coactivator CGI-58 (-25%, F30 vs Ctrl, p<0.05), and the increased expression of its inhibitor G0S2 (+71%, F30 vs Ctrl, p<0.05). Therefore, it is possible that the interaction of the above-mentioned ATGL regulatory proteins affected its (ATGL) mRNA expression in high intensity group.
3. CGI-58 is positive regulator of ATGL and lipolysis. You found CGI-58 mRNA is decreasing in left ventricles after 30 minute moderate and intense exercise. How is it possible?
The lack of exercise-induced changes in CGI-58 protein content was accompanied by a decrease in its transcript level in the left ventricle after 30 minutes of moderate and intense exercise. Accordingly, cardiac ATGL mRNA expression was significantly reduced in F30 group, and unchanged in M30 group. These data suggest that the acute exercise model (in the left ventricle) is characterized by stable protein expression of the main compounds of the lipolytic system. This likely stems from the relatively short and acute form of exercise. Therefore, we can speculate that the left ventricle of the heart was not subjected to extraordinary metabolic challenges that would require pronounced increases in energetic substrates turnover during a single bout of physical activity. It is hard to properly discuss the topic, given that to the best of our knowledge the regulation of mRNA CGI-58 expression in the myocardium under physiological conditions remains largely unexplored. Since it occurred only in left ventricle we suspect that rather a set of local factors was responsible for the phenomenon.
4. Similarly you found increase in mRNA level of G0S2 which is negative regulator of lipolysis. Can you give explanation for that?
The increased mRNA level of G0S2 in the running rats is consistent with the observed stable protein content of G0S2 in left ventricle. The observed changes, however, are difficult to explain. Transcriptional control (the amount of mRNA) is often not readily evident, as there are numerous observations show a poor relationship between mRNAs and protein content. In some circumstances, a better (more direct) approach is to observe changes on the post-transcriptional (protein expression) level. This was noticed not only in cancer cells, but also in fully differentiated, healthy cells and tissues [Knapp P, et al., Prostaglandins Other Lipid Mediat. 2016 Mar;123:9-15, Knapp P, et al., Horm Metab Res. 2012 Jun;44(6):436-41, Harasiuk D, et al., Cell Physiol Biochem. 2015;35(3):1095-106, Baranowski M., Cell Physiol Biochem. 2014; 33(4):1047-57, Chabowski A, et al., Naunyn Schmiedebergs Arch Pharmacol. 2006 Jul;373(4):259-63]. These observations suggest that, rather than transcriptional modifications, the alteration of the lipolytic system are likely to be more dependent on the changes in the post-translational modification of the compounds.
5. You found increase in FFA level in plasma but not in left ventricles of heart, does it mean heart tissues keep using glucose as a source of energy after exercise?
It has been observed that acute exercise, i.e. a single bout that lasts from several minutes to hours, results in an abrupt increase in glucose uptake and oxidation [PMID: 11009433, PMID: 3198763]. Moreover, a significant portion of the myocardial glucose is derived from endogenous glycogen stores. If exercise intensity is moderate enough and continued for an extended time period a greater percentage of fatty acid is used [PMID: 23395166]. This coincides with elevated plasma free fatty acid concentrations due to enhanced adipose tissue lipolysis [PMID: 10919960]. Our findings show that increased intensity (F30 group) and duration (M120) of the physical activity results in a decreased fatty acids content (in particular lipids like DG, TG) partly due to an elevated mRNA expression of ATGL and its lipolytic activity in the right ventricle. These differences do not appear after 30 minutes of moderate intensity run (M30 group). Therefore, we may postulate that the ratio of glucose to fatty acids usage during acute exercise is increased (greater glucose consumption) with the increasing intensity and duration of the exercise. On the other hand, it is also possible that during physical activity an increased plasma FFA concentration elevated myocardial lipids level. The lipids might have been subsequently used as an energy source for working heart muscle, thus saving the heart’s internal storage. This would result in no observable changes in the lipid pool of left ventricle. Based on the obtained results we would postulate the second mechanism as the one more likely.
6. ‘Two hours run causes a significant drop in the total DG content’ (line 154). How is it possible if total FFA level is not changing?
Please notice that total FFA content in left ventricle was increased by 36% (M120 vs Ctrl), but did not reach a significance level. Interestingly, DG level was diminished to a similar extent (-28%, M120 vs Ctrl, p<0.05). These results would also suggest a relationship between the DG hydrolysis and intracellular FFA availability.
7. Why did only F30 group show increase CGI-58 expression in right ventricle but no other exercise group?
In line with the increased mRNA expression of CGI-58, high intensity run markedly elevated ATGL transcript level (+1,3 fold, F30 vs Ctrl, p<0.05) in right ventricle. Consistently, F30 group exhibited a significantly lower lipid contents in all the investigated fractions namely: FFA, DG and TG (-21%, -33%, -26% respectively, F30 vs Ctrl, p<0.05). Thus, we can speculate that fast speed of the run (28 m/min, which is equal to approximately 82% of the maximal oxygen uptake, PMID: 944429) induced substantial changes in the rate of lipolysis compared with the moderate speed of the run (18 m/min, i.e., approximately 65% of the maximal oxygen uptake, PMID: 944429) at both durations.
8. Why did long exercise cause (120 minutes) decrease in CGI-58 protein level? Should it not increase?
Moderately intensely running group (M120) exhibited lower protein expression of CGI-58 in the right ventricle, however this drop was statistically significant only when compared with short, moderately running rats (M30). Still, it is difficult to satisfactorily explain this phenomena, since we noticed an increased total lipolytic activity and a decreased TG content in M120 group (M120 vs M30).
9. 'General schema of the conducted research' (line 379) is hard to understand and it is not fully explanatory. Either remove it or make more explanatory.
Thank You for the suggestion. Prompted by Your (and Reviewer 2) suggestion we removed (actually replaced it with barplots) figure no. 6 from the manuscript.
10. Why treadmill was inclined at 10°?
Running is a common physical activity. The sloping planes upon which individuals run can be classified as: uphill running (UR), level running (LR), and downhill running (DR). These scenarios differ regarding muscle contraction patterns and physiological workloads. Physiological variables such as energy expenditures, heart rate, and ratings of perceived exertion (RPE) responses are greater during UR compared to DR or LR [PMID: 9302491]. Treadmill inclination is often used to increase work-load (thus simulating uphill running). In our protocol we used +10o incline to mimic the experiment performed by Lawler et al. (PMID: 8289613) in which oxygen consumption was measured at those two different speeds.
11. PCR condition is not described in method.
Thank You for spotting this drawback. We have added the missing information into the material and methods section of the manuscript.
12. Catalogue number for antibodies is missing in method section.
Once again, thank You for the scrupulous reading of our manuscript and for pointing that out. The catalogue numbers for antibodies have been added to the material and methods section of the manuscript.
Reviewer 2 Report
Reviewer’s comments and suggestions for Authors
In this research manuscript, the authors have evaluated the time and intensity dependent effects of exercise/physical activity on the heart components of the lipolytic complex. For this research, the authors used Wistar rats that run on a treadmill with two different speed and duration. The study has measured the mRNA and protein expressions of ATGL– adipose triglyceride lipase, CGI-58– comparative gene identification-58, G0S2– G0/G1 switch gene 2 and HSL– hormone-sensitive lipase. They also measured the biochemical levels of lipid content. The research reported that there has been no changes in left ventricle lipid contents and only minor alteration in its ATGL mRNA levels. Authors reported a difference with its right counterpart. In conclusion, the author has revealed that exercise affects the enzymatic and biochemical components of the lipolytic system and the heart ventricles.
The manuscript is well written in terms of the English language. However, there few comments and advise to modify in the current version of the manuscript and submit again as a revised version.
1. In the abstract part of the paper, some corrections are needed to incorporate. Such as rt should be changed with real-time, Blot should be blot, the lipid content should not be abbreviated as they are used the first time. Full form of GLC should be written.
2. As in your previous published study based on a tachycardia (increase in the heart beat) your observation was different from this study. Please explain the contraction. However, I understand that in this study the authors had reported the single bout of treadmill run. My other question is how was the author regulate the speed of the animals. Did they train them for the experiment?
3. What was the reason for the observation the control group, we found an increased expression of mRNA for ATGL in M120 group (+83%, MR_120 min vs. Ctrl, p < 0.05, Fig. 1A), whereas a decrease was noticed for ATGL mRNA level in F30 (-25%, F30 vs. Ctrl, p < 0.05, Fig. 1A)” please specify.
4. “Total protein expression of ATGL, CGI-58, and G0S2 seemed to be virtually constant among all the studied groups “is there is any specific reason for that. Please discuss in the result in the discussion.
5. The result of western blot needs to change based on your real figure “Figure 1. Effects of treadmill running on the mRNA and protein expressions of ATGL (A, E), CGI-58(B, F), G0S2 (C, G), HSL (D, H) in left ventricle”. Try to increase the resolution of your pictures.
6. Please explain the reason for peculiar result of the total lipid content in the left ventricle in the result section as it will help the reader to understand properly.
7. I could not understand the reason for the figures results (Fig. 4E, 4G, and 4H). Why the observation is different in this ventricles.
8. Page number 13, the authors have used reference 15 to explain their study, however, those results were on type 2 diabetes. The behaviors of human types 2 diabetics are different compared to the normal rat used in the study. I hope the author has to search for another previously published study used as a reference.
9. Page number 14, the authors have to discuss the results of the important studies such as references 37 and 38.
10. The line need to have a reference “It is likely that cardiomyocytes of both ventricles were exposed to the same humoral and neural changes during exercise and thus those factors should not contribute to the differences between the ventricles”.
11. In the conclusion part, the author mentioned that “We suggest a possible role of intrinsic factors in the regulation of endogenous TG lipolysis in each ventricle”. But they need to explain the behavior in right ventricles properly.
12. In the animal experiment section, please write the ethical approval number in the revised version.
13. Figure 6, need to be properly illustrated and redraw by the authors.
14. Please check the references 18 and 41, not formatted based on the author guidelines.
Author Response
Dear Reviewer 2,
Thank You for an in-depth analysis of our study and constructive comments. We appreciate your work and time spent on reviewing the manuscript. Below You will find our answers to the discussed issues. For Your greater convenience we have placed them (as well as all the changes in the manuscript) with red font. We hope that You will find them satisfactory.
Best regards, Agnieszka Miklosz – corresponding author
1. In the abstract part of the paper, some corrections are needed to incorporate. Such as rt should be changed with real-time, Blot should be blot, the lipid content should not be abbreviated as they are used the first time. Full form of GLC should be written.
Thank You for the suggestion. Of course, we agree. The indicated issues have been corrected.
2. As in your previous published study based on a tachycardia (increase in the heart beat) your observation was different from this study. Please explain the contraction. However, I understand that in this study the authors had reported the single bout of treadmill run. My other question is how was the author regulate the speed of the animals. Did they train them for the experiment?
The rats were made to run on an electrically motor-driven treadmill which speed could be easily regulated. In our study, as described in the materials and methods section, all animals were first subjected to a preliminary treadmill adjustment. Namely, they run 15 min daily for five days with the same speed as in the proper experiment. Afterwards the rats were randomly assigned to their respective groups. In the case of the training groups the variables such as time, speed and inclination of the treadmill were closely monitored.
Regarding the previous study on tachycardia, the heart rate was accelerated by electrical stimulation (pacemaker). The rate of stimulation was 600/min and the stimuli were followed by contractions as evidenced by recording of ECG. The limit of 600/min was the maximal one in that respect that each stimulus was followed by contractions. Above that rate of stimulation the contractions did not always follow the rate of stimulation. In the present experiment the rate of contractions was regulated “naturally” according to the need for oxygen uptake. Certainly, the heart rate they did not reach 600/min. The treadmill exercise affected the whole body as well as the heart itself. Therefore, the environmental conditions in the two experiments were basically different. The differences concerned both the metabolic and hormonal conditions in the two experiments.
3. What was the reason for the observation the control group, we found an increased expression of mRNA for ATGL in M120 group (+83%, MR_120 min vs. Ctrl, p < 0.05, Fig. 1A), whereas a decrease was noticed for ATGL mRNA level in F30 (-25%, F30 vs. Ctrl, p < 0.05, Fig. 1A)” please specify.
All animals (also those form the control group) underwent a preliminary treadmill adjustment by running 15 min daily for five days with the same speed as in the proper experiment. Knowing the fact that even such an adjustment training itself can cause changes in lipid metabolism in the heart and in blood plasma, all rats were familiarized with treadmill in the same manner, so that we could ‘nullified’ this effect in our study. Therefore, firstly we compared the results to the untrained rats (control group), and then between the studied groups. Furthermore, we have to stress out that a preliminary treadmill adjustment is frequently used in the exercise literature’s training protocols.
4. “Total protein expression of ATGL, CGI-58, and G0S2 seemed to be virtually constant among all the studied groups “is there is any specific reason for that. Please discuss in the result in the discussion.
A good question. In left ventricle the relatively stable protein content (ATGL, CGI-58, G0S2) was in agreement with the unchanged total lipolytic activity as well as lipids content. This coincides with the increased plasma free fatty acid concentrations, which we suspect was sufficient to cover the increased demand for lipid utilization in working cardiomyocytes. On the other hand, one cannot rule out an increased glycolysis, especially when energy demands are increased by exercises in the heart where oxygen supply is highly depended exclusively on increased blood supply by coronary system. On the other hand, it is also possible that the left ventricle of the heart was not subjected to extraordinary metabolic challenges that might evoke pronounced increases in energy turnover during single bout of physical exertion.
5. The result of western blot needs to change based on your real figure “Figure 1. Effects of treadmill running on the mRNA and protein expressions of ATGL (A, E), CGI-58(B, F), G0S2 (C, G), HSL (D, H) in left ventricle”. Try to increase the resolution of your pictures.
Thank You for the comment. We have corrected the figures and we hope that in this form they will Your expectations.
6. Please explain the reason for peculiar result of the total lipid content in the left ventricle in the result section as it will help the reader to understand properly.
We are aware that the stable total lipid content in the left ventricle, in contrast to the right one, seems to be rather odd. In the literature, the systemic usage of glucose and fatty acids in response to acute exercise has been well established by measuring the respiratory quotient. To provide the heart with a sufficient amount of energy during physical exercise, a switch in energy substrates occurs from an initial preference for glucose to fatty acids. Along this line, it has been observed that acute exercise, i.e. a single bout typically lasting from several minutes to hours, results in an abrupt increase in glucose uptake and oxidation [PMID: 11009433, PMID: 3198763]. If exercise intensity is moderate enough and continiued for an extended time period a greater percentage of fatty acids is used [PMID: 23395166]. This coincides with increased plasma free fatty acid concentrations due to enhanced white adipose tissue lipolysis [PMID: 10919960]. In this study, together with an increased FA mobilization from adipose tissue, we noticed unchanged lipids content in the left ventricle after acute treadmill run. Taken altogether, the unchanged TG content and unaffected total lipolytic activity indicate that the increased circulating FA levels themselves were sufficient to cover the increased demand for lipid utilization in working cardiomyocytes.
7. I could not understand the reason for the figures results (Fig. 4E, 4G, and 4H). Why the observation is different in this ventricles.
Thank you for rising this point. We agree that protein content of the examined compounds of the lipolytic system (ATGL, G0S2 and HSL) was relatively unchanged in both ventricles of the heart. Nevertheless, we observed virtually no changes in left ventricle lipid contents and only minor fluctuations in its ATGL mRNA levels. This was in contrast with its right counterpart i.e. the content of TG and DG decreased in response to both increased duration and intensity of a run. This occurred in tandem with increased mRNA expression for: ATGL, CGI-58 and decreased expression of G0S2. To sum up, changes observed in the left ventricle did not mirror those in the right one, apart from the protein content of the main compounds of the lipolytic system.
8. Page number 13, the authors have used reference 15 to explain their study, however, those results were on type 2 diabetes. The behaviors of human types 2 diabetics are different compared to the normal rat used in the study. I hope the author has to search for another previously published study used as a reference.
Thank You for spotting this mistake. We have removed the reference 15 and 30 (a study on people with obesity and T2DM) and now we rely only on the data from rodents.
9. Page number 14, the authors have to discuss the results of the important studies such as references 37 and 38.
In response to Your remark we have discussed the above-mentioned studies in our manuscript.
10. The line need to have a reference “It is likely that cardiomyocytes of both ventricles were exposed to the same humoral and neural changes during exercise and thus those factors should not contribute to the differences between the ventricles”.
Sorry, but we are not aware of the existence of such a reference. Now we changed the above-mentioned sentence on: “Both ventricles were perfused with the arterial blood and therefore were exposed for the same blood-borne factors [PMID: 28984631]. It is a good reason to presume that branches of the autonomic nervous system innervating both ventricles provided the same impulses to each ventricle to enable them to work in concert. Therefore, those factors could not be responsible for the differences in fat metabolism between the two ventricles in response to exercise”.
11. In the conclusion part, the author mentioned that “We suggest a possible role of intrinsic factors in the regulation of endogenous TG lipolysis in each ventricle”. But they need to explain the behavior in right ventricles properly.
The right ventricle muscle is much thinner and develops much lower systolic pressure than the left ventricle one. Therefore, the conditions for generating and operating of the hypothesized local factors are different the ventricles. It might reflect on the differences in behavior of the examined lipolytic system compounds and fat metabolism between the ventricles.
12. In the animal experiment section, please write the ethical approval number in the revised version.
Now, we have added permission number in the manuscript.
13. Figure 6, need to be properly illustrated and redraw by the authors.
Prompted by Your suggestion and Reviewer 1 we removed figure no. 6 from the manuscript.
14. Please check the references 18 and 41, not formatted based on the author guidelines.
Thank You for spotting this mistake. We have formatted the indicated references according Journal requirements.
Reviewer 3 Report
Article by Miklosz et al,. describes the observation on the lipolytic response pattern between the right and left ventricle of heart component after treadmill running at different time points using wistar rats. They found out that right ventricle lipolytic complex was significantly affected as compared to left ventricle as proved by quantifying the mRNA and protein expression pattern of key genes/proteins involved in lipolysis, in addition to lipid content.
There are number of other genes that are involved in heart function and regulated during exercise such as PPAR-α and PPAR-δ target genes which need to be checked.
Are there any histological changes observed before and after exercise? This can be shown by H and E staining of the heart sections.
Western blotting experiment of CGI-58 and HSL are not clear. Please provide better image in the revised version including GAPDH.
GOS2 expression in fig 4 is not the representative figure of quantified data.
Author Response
Dear Reviewer 3,
Thank You for Your valuable comments that have contributed to the improvement of the manuscript quality. Below You will find our answers/explanations, written in a point-by-point manner. For Your greater convenience we have placed all the changes in the manuscript with the red color. We hope that You will find them satisfactory.
Best regards,
Agnieszka Mikłosz – corresponding author
1. There are number of other genes that are involved in heart function and regulated during exercise such as PPAR-α and PPAR-δ target genes which need to be checked.
Thank You for rising this concern. Unfortunately, given the time constraints (10 days for response) we couldn’t procure desired primers/antibodies. We did, however examined the expression of PGC-1alpha, i.e. PPARs’ coactivator. We trust that the results obtained for its mRNA and protein levels will be satisfactory and in line with Your expectations. Of course, the outcomes of the analysis were placed in the results and discussion sections.
2. Are there any histological changes observed before and after exercise? This can be shown by H and E staining of the heart sections.
Unfortunately, we did not perform histological analysis of cardiomyocytes from left and right ventricles (H & E staining) before and after single bout of treadmill running. We are sorry for this inconvenience. We are aware that cardiac morphology can change during physical exercise, however since our project has been completed we do not have time (because of 10 days for review process) to set another one. Moreover, performing such analysis would require repeating the experminents on new rats. This would require another consent of the ethical board for animal care.
3. Western blotting experiment of CGI-58 and HSL are not clear. Please provide better image in the revised version including GAPDH.
Now, we have corrected the figures. And we hope that in this form they will fulfill the Journal standards.
4. G0S2 expression in fig 4 is not the representative figure of quantified data.
Thank You for the comment. We have corrected the figure and we hope that in this form they will Your expectations.
Round 2
Reviewer 1 Report
author made all improvements suggested in previous review. No change required now.
Reviewer 3 Report
No comments.